# Polymorphisms of *FDPS*, *LRP5*, *SOST* and *VKORC1* genes and their relation with osteoporosis in postmenopausal Romanian women

**Alina Deniza Ciubean**[1], **Rodica Ana Ungur**[1]*, **Laszlo Irsay**[1]*, **Viorela Mihaela Ciortea**[1], **Ileana Monica Borda**[1], **Gabriela Bombonica Dogaru**[1], **Adrian Pavel Trifa**[2], **Stefan Cristian Vesa**[3], **Anca Dana Buzoianu**[3]

1 Department of Rehabilitation, University of Medicine and Pharmacy "Iuliu Haţieganu", Cluj-Napoca, Romania, 2 Department of Genetics, University of Medicine and Pharmacy "Iuliu Haţieganu", Cluj-Napoca, Romania, 3 Department of Pharmacology, Toxicology and Clinical Pharmacology, University of Medicine and Pharmacy "Iuliu Haţieganu", Cluj-Napoca, Romania

* rodica.ana.ungur@gmail.com (RAU); irsaylaszlo@gmail.com (LI)

**Data Availability Statement:** All relevant data are within the paper and its Supporting Information file.

## Abstract

### Objectives

This study aimed to assess the relationship between bone mineral density and genotypes of four polymorphisms in previously detected osteoporosis-candidate genes (*FDPS* rs2297480, *LRP5* rs3736228, *SOST* rs1234612, *VKORC1* rs9934438) in postmenopausal Romanian women with primary osteoporosis.

### Methods

An analytical, prospective, transversal, observational, case-control study on 364 postmenopausal Romanian women was carried out between June 2016 and August 2017 in Cluj Napoca, Romania. Clinical data and blood samples were collected from all study participants. Four polymorphisms were genotyped using TaqMan SNP Genotyping assays, run on a QuantStudio 3 real-time PCR machine.

### Results

Women with TT genotype in *FDPS* rs2297480 had significantly lower bone mineral density values in the lumbar spine and total hip, and the presence of the T allele was significantly associated with the osteoporosis. Women carrying the CC genotype in *LRP5* rs3736228 tend to have lower bone mineral density values in the femoral neck and total hip. No significant association was found for the genotypes of *SOST* rs1234612 or *VKORC1* rs9934438.

### Conclusions

Our study showed a strong association between bone mineral density and polymorphisms in the *FDPS* gene, and a borderline association with *LRP5* and *SOST* polymorphisms in

**Funding:** The authors received no specific funding for this work.

**Competing interests:** The authors have declared that no competing interests exist.

postmenopausal Romanian women with osteoporosis. No association was found for *VKORC1*.

## Introduction

Osteoporosis (OP) is a common and complex skeletal disorder characterized by decreased bone mass and bone quality, which lead to an increase vulnerability to fragility fractures [1]. From a genetic perspective, the etiology of OP and fracture risk susceptibility is multifactorial, involving significant environmental influence together with genetic factors across numerous biologic processes, and it is thought that 60% to 80% of bone loss acceleration is due to genetic factors. [2]

More than 66 bone mineral density (BMD) loci have been studied in genome wide association studies (GWASs), confirming the highly polygenic nature of BMD variation. Although over the past decades there has been significant progress in identifying candidate genes involved in BMD, fracture and other related phenotypes, most of the genetic variants remain to be uncovered or validated in various ethnic groups.

To date, multiple single nucleotide polymorphisms (SNPs) in several genes have been associated to BMD, but the results are inconclusive and conflicting. As OP is a polygenic disease, and each bone phenotype (density, quality, metabolic rate) is the result of interaction among multiple genes, and the "essential" one, responsible for OP, is not yet identified despite using advanced methods [3,4].

One of the most important signaling pathways in bone in Wnt, which is crucial for bone development during embryogenesis and has a dual role in bone mass regulation, influencing both bone formation and resorption. The components of the Wnt pathway are proteins involved in cell proliferation, differentiation, and apoptosis of bone cells.[5] When cells are stimulated through the membrane receptors low-density lipoprotein receptor-related protein 5/6 (LRP5/6), the architecture of the multiprotein complex is modified, which inhibits b-catenin, leading to its translocation to the nucleus, where it initiates the transcription of target genes.[6] *LRP5* is the most important membrane receptor of the Wnt signaling pathway and it was previously tagged in a genome-wide association study to be associated with OP.[7] *LRP5* inactivation caused by mutation is responsible for osteoporosis-pseudoglioma syndrome, in which low bone mass and fractures occur.[8] Also, there are several naturally occurring inhibitors of Wnt signaling, such as Dickkopf (DKK) and sclerostin (SOST) proteins that inactivate signaling from LRP5/6 receptors. Sclerostin, encoded by the SOST gene, antagonizes Wnt signaling in both osteocytes and osteoblasts by binding to the LRP5/6 coreceptor and preventing bone formation.[9,10] High-bone-mass syndromes are thought to be caused by inactivating mutations of SOST (sclerosteosis and van Buchem's disease).[11] Animal studies have indicated that sclerostin inhibition increases bone mass by stimulating bone formation and inhibiting bone resorption.[12]

The discovery of mevalonate pathway's role as the target of the antiresorbtive agents from the amino-bisphosphonates (N-BP) class, used in postmenopausal OP treatment, revealed other important genes to be considered as candidate genes, like farnesyl diphosphate synthase (*FDPS*) or geranylgeranyl pyrophosphate synthase (*GGPS1*), which maintain the resorption activity of the osteoclasts. *FDPS* is a key-enzyme of the mevalonate pathway and is a well recognised target of several N-BPs, making it worthy of being studied. [13–19]

Also, even though its hemostatic effect and implication in warfarin sensitivity are well known, there has been evidence that vitamin K also plays an important role in maintaining bone strength and that mutations in the vitamin K epoxy reductase (*VKORC1*) gene may modify the gamma-carboxylation of osteocalcin and may influence BMD [20–22].

Today, it is well established that OP is a multifactorial complex disorder, whose pathogenesis is due to the interaction of various genetic determinants regulating bone and mineral metabolism with "non-skeletal" risk factors that could influence fall risk (e.g. muscle strength, balance, visual acuity), environmental influences and lifestyle [23]. However, to date no gene has been definitely identified as a major gene for OP.

The purpose of the present study is to evaluate the relation between BMD and genotypes of four SNPs in previously reported osteoporosis candidate-genes (*FDPS* rs2297480, *LRP5* rs3736228, *SOST* rs1234612, *VKORC1* rs9934438) in a cohort of postmenopausal Romanian women.

## Material and methods

### Study population

An analytical, prospective, transversal, observational, case-control study on 364 postmenopausal Romanian women was carried out between June 2016 and August 2017. All the women included in the study were recruited either from the inpatient clinic or during routine outpatient visit in the Clinical Rehabilitation Hospital in Cluj-Napoca, Romania.

The inclusion criteria was as follows: women, at least 45 years old, that had been menopausal for at least 2 years and had a recent diagnosis of osteoporosis or normal density result on BMD measurement using dual-energy X-ray absorbtiometry (DEXA). The women included in the study were divided into two groups: osteoporosis (n = 228) and healthy age-matched controls (n = 136). Patients with history of metabolic bone diseases (e.g. hyperparathyroidism, osteomalacia, Paget disease), malignancy, bone metastasis and those treated with drugs that influence bone metabolism (e.g. anti-osteoporotic drugs or vitamin K antagonists) were excluded from the study.

Clinical data was collected by interview and from the medical documents of each patient: age, body mass index, years since menopause, lumbar spine (L1-L4), femoral neck and total hip BMD. Furthermore, a 2 ml EDTA vial of peripheral blood was collected from each study participant for the genetic testing.

The study was approved by the Ethics Committee of the University of Medicine and Pharmacy "Iuliu Hațieganu" Cluj-Napoca (approval no. 248/09.06.2016). All participants were informed of the characteristics of the study and all gave signed informed consent regarding the genetic testing and clinical data collection prior to inclusion.

### SNP genotyping

Genomic DNA was obtained from peripheral blood withdrawn on EDTA, using commercially available kits (Quick gDNA MiniPrep kit, Zymo Research, USA; PureLink Genomic DNA Mini Kit, Invitrogen, Thermo Fisher, USA). We genotyped four SNPs (FDPS rs2297480, LRP5 rs3736228, SOST rs1234612, VKORC1 rs9934438) in all patients and controls using the real-time PCR technique. Standard, predesigned TaqMan SNP genotyping assays, containing all the primers and probes needed for genotyping, were purchased from Thermo Fisher (codes C___2737970_10, C__25752205_10, C___7566033_10, C__30204875_10). All the genotyping were performed according to manufacturer's instructions. The reaction mix contained 10 μl of 2xTaqMan Genotyping Master Mix (Applied Biosystems, Thermo Fisher, USA), 0.5 μl of the corresponding 40xTaqMan SNP genotyping assay, approximately 25 ng of genomic DNA and

free-nucleases water to the final volume of 20 μl. The same amplification program was used for all the genotyping, consisting in a pre-read stage of 30 seconds (s) at 60˚C, hold stage of 10 minutes (min) at 95˚C, followed by the PCR stage, consisting of 40 cycles, each comprising 15 s at 95˚C and 1 min at 60˚C, and a post-read stage of 30 s at 60˚C. All the experiments were run on a QuantStudio 3 real-time PCR machine (Applied Biosystems, Thermo Fisher, USA).

## Statistical analysis

Statistical analysis was performed using MedCalc Statistical Software version 18.6 (MedCalc Software bvba, Ostend, Belgium; http://www.medcalc.org; 2018). Qualitative data were characterized by frequency and percent. Quantitative data were described by mean and standard deviation or median and 25–75 percentiles, depending on the normality of distribution at Kolmogorov Smirnov test. Differences between groups were tested using the chi-square or Mann-Whitney, whenever appropriate (t-test, Fischer). A p-value <0.05 was considered statistically significant. The size sample estimation was calculated by assessing the distribution of the TT genotype for FDPS rs2297480 SNP in a group of 23 patients (52.2%) with osteoporosis and 13 controls (38.5%). For a type I error of 0.05 and a type II error of 0.20, we calculated a number of 162 controls and 290 patients.

## Results

A total of 364 patients met the inclusion criteria and were included in the study to be genotyped, of which 228 had a diagnosis of OP and were compared to 136 age-matched healthy controls. The mean age in the osteoporosis group was 65.5 years (±7.39), and 63.45 (± 8.16) in the control group, respectively. The osteoporosis group had signficantly more years of amenorrhea than the control group, lower BMD values at all measured sites and a higher fracture risk (all p<0.05).

The main characteristics of the women included in the study are shown in Table 1.

The distribution of the *FDPS* rs2297480, *LRP5* rs3736228, *SOST* rs1234612 and *VKORC1* rs9934438 genotypes and allele frequencies in the study population (n = 364) are shown in Table 2.

**Table 1. Clinical characteristics of women included in the study.**

| Variables | Osteoporosis *n* = 228 | Controls *n* = 136 | *p*-value |
|---|---|---|---|
| Age, mean ± SD [years] | 65.5 ± 7.39 | 63.45 ± 8.16 | **0.014** |
| BMI, mean ± SD [kg/m$^2$] | 27.05 ± 4.74 | 30.56 ± 5.40 | **< 0.001** |
| Age at menopause, mean ± SD [years] | 47.26 ± 4.84 | 48.35 ± 4.88 | **0.045** |
| Time of amenorrhea, mean ± SD [years] | 18.25 ± 8.36 | 15.08 ± 8.82 | **0.001** |
| Previous fragility fracture (vertebral, hip, wrist, humerus), *n* (%) | 132 (57.9) | 0 (0) | - |
| Current smoking, *n* (%) | 10 (4.4) | 6 (4.4) | 0.991 |
| Alcohol consumption > 3 units/day, *n* (%) | 3 (1.3) | 0 (0) | - |
| Parent fractured hip, *n* (%) | 13 (5.6) | 8 (5.8) | 0.943 |
| Lumbar spine (L1-L4) BMD, mean ± SD [g/cm$^2$] | 0.851 ± 0.11 | 1.116 ± 0.15 | **< 0.001** |
| Femoral neck BMD, mean ± SD [g/cm$^2$] | 0.751 ± 0.10 | 0.969 ± 0.21 | **< 0.001** |
| Total hip BMD, mean ± SD [g/cm$^2$] | 0.791 ± 0.5 | 0.968 ± 0.21 | **<0.001** |
| FRAX—10 year risk of major osteoporotic fracture, mean ± SD [%] | 8.04 ± 4.68 | 4.34 ± 2.47 | **0.005** |
| FRAX—10 year risk of hip fracture, mean ± SD [%] | 2.76 ± 2.97 | 0.82 ± 1.47 | **< 0.001** |

**Table 2. Allelic and genotypic frequencies of the SNPs in the study population (n = 364).**

| Gene | SNP | Genotypes (%) | | | Allele (%) | |
|------|-----|------|------|------|------|------|
| *FPDS* | rs2297480 | **TT** | **GT** | **GG** | **T** | **G** |
| | | 222 (60.99) | 124 (34.07) | 18 (4.95) | 568 (78) | 160 (22) |
| *LRP5* | rs3736228 | **CC** | **CT** | **TT** | **C** | **T** |
| | | 257 (70.6) | 103 (28.3) | 4 (1.1) | 617 (85) | 111 (15) |
| *SOST* | rs1234612 | **TT** | **CT** | **CC** | **T** | **C** |
| | | 183 (50.27) | 142 (39.01) | 39 (10.71) | 508 (70) | 220 (30) |
| *VKORC1* | rs9934438 | **GG** | **AG** | **AA** | **G** | **A** |
| | | 126 (34.62) | 156 (42.86) | 82 (22.53) | 408 (56) | 320 (44) |

Women carrying the TT genotype of *FDPS* rs2297480 have significantly lower BMD values in the lumbar spine and total hip than the heterozygous GT or homozygous GG (p = 0.006 and p = 0.03, respectively), but not in the femoral neck (p = 0.179) (Table 3).

After dividing the patients into groups, analysis of the codominant model was performed. Codominant models hypothesize that the major allele homozygotes, the heterozygotes or the minor allele homozygotes are associated with the lowest, the intermediate, and the highest risk, respectively, or they are associated with the highest, the intermediate, and the lowest risk, respectively.

Results show that when analyzing the codominant model, the TT genotype of rs2297480 SNP continues to be significantly associated with OP risk (OR = 2.1; 95% CI = 1.4–3.3; p-value<0.05), while the GT genotype had the lowest risk (OR = 0.4; 95% CI = 0.3–0.7; p-value = 0.001). Furthermore, the presence of the T allele was significantly associated with OP in the group analysis (p = 0.005) (Table 4).

As for *LRP5* rs3736228 SNP, the homozygous CC have significantly lower BMD values in the femoral neck and total hip than the CT or TT genotypes (p = 0.028 and p = 0.014), but not in the lumbar spine region (p = 0.717) (Table 3). In the codominant model of the groups analysis, this association continues to reach statistical significance in the OP group (OR = 1.5; 95%

**Table 3. Bone mineral density values among genotypes (n = 364).**

| Gene (SNP) | Genotype | BMD L1-L4 g/cm² (IQR) | BMD FEMORAL NECK g/cm² (IQR) | BMD TOTAL HIP g/cm² (IQR) |
|------------|----------|------------------------|-------------------------------|----------------------------|
| *FDPS* rs2297480 | GG | 0.910 (0.804; 1.023) | 0.840 (0.699; 0.925) | 0.865 (0.747; 1.030) |
| | GT | 0.981 (0.852; 1.102) | 0.803 (0.725; 0.885) | 0.882 (0.779; 0.997) |
| | TT | 0.865 (0.778; 1.091) | 0.774 (0.705; 0.873) | 0.833 (0.746; 0.939) |
| | p-value | **0.006** | 0.179 | **0.030** |
| *LRP5* rs3736228 | TT | 0.877 (0.669; 1.053) | 0.782 (0.711; 0.861) | 0.832 (0.756; 0.945) |
| | CT | 0.943 (0.788; 1.099) | 0.825 (0.714; 0.918) | 0.895 (0,771; 1.019) |
| | CC | 0.924 (0.836; 1.061) | 0.715 (0.635; 0.747) | 0.810 (0.644; 0.882) |
| | p-value | 0.717 | **0.028** | **0.014** |
| *SOST* rs1234612 | CC | 1.004 (0.897; 1.085) | 0.792 (0.734; 0.891) | 0.896 (0.742; 1.008) |
| | CT | 0.941 (0.838; 1.076) | 0.787 (0.702; 0.870) | 0.850 (0.759; 0.950) |
| | TT | 0.910 (0.812; 1.046) | 0.799 (0.714; 0.886) | 0.837 (0.757; 0.965) |
| | p-value | 0.124 | 0.596 | 0.679 |
| *VKORC1* rs9934438 | AA | 0.959 (0.848; 1.120) | 0.803 (0.711; 0.931) | 0.860 (0.770; 1.019) |
| | AG | 0.910 (0.812; 1.055) | 0.792 (0.702; 0.870) | 0.832 (0.742; 0.948) |
| | GG | 0.926 (0.823; 1.042) | 0.794 (0.721; 0.873) | 0.847 (0.771; 0.963) |
| | p-value | 0.086 | 0.367 | 0.133 |

**Table 4. Allelic and genotypic distribution between osteoporotic women and controls.**

| Gene | Polymorphism | Model/alleles | Genotypes, alleles | Osteoporosis (%) (n = 228) | Controls (%) n = 136 | OR [95% CI] | p-value |
|------|--------------|---------------|--------------------|-----------|-----------|-------------|---------|
| FDPS | rs2297480 | Codominant | GG | 10 (4.4) | 8 (5.9) | 0.7 [0.2–1.9] | 0.524 |
| | | | GT | 63 (27.6) | 61 (44.9) | 0.4 [0.3–0.7] | **0.001** |
| | | | TT | 155 (68) | 67 (49.3) | 2.1 [1.4–3.3] | **<0.05** |
| | | Major allele | T | 373 (81.8) | 195 (71.7) | - | **0.005** |
| | | Minor allele | G | 83 (18.2) | 77 (28.3) | - | 0.127 |
| LRP5 | rs3736228 | Codominant | CC | 169 (74.1) | 88 (64.7) | 1.5 [0.9–2.4] | **0.05** |
| | | | CT | 56 (24.6) | 47 (34.6) | 0.6 [0.3–0.9] 0.6 [03.-0.9] | **0.041** |
| | | | TT | 3 (1.3) | 1 (0.7) | 1.8 [0.1–17.4] | 0.612 |
| | | Major allele | C | 394 (86.4) | 223 (82) | - | 0.143 |
| | | Minor allele | T | 62 (13.6) | 49 (18) | - | 0.527 |
| SOST | rs1234612 | Codominant | CC | 19 (8.3) | 20 (14.7) | 0.5 [0.2–1] | 0.057 |
| | | | CT | 88 (38.6) | 54 (39.7) | 0.9 [0.6–1.4] | 0.834 |
| | | | TT | 121 (53.1) | 62 (45.6) | 1.3 [0.8–2] | 0.167 |
| | | Major allele | T | 330 (72.4) | 178 (65.4) | - | 0.101 |
| | | Minor allele | C | 126 (27.6) | 94 (34.6) | - | 0.266 |
| VKORC1 | rs9934438 | Codominant | AA | 46 (20.2) | 36 (26.5) | 0.7 [0.4–1.1] | 0.164 |
| | | | AG | 101 (44.3) | 55 (40.4) | 1.1 [0.7–1.8] | 0.472 |
| | | | GG | 81 (35.5) | 45 (33.1) | 1.1 [0.7–1.7] | 0.636 |
| | | Major allele | G | 263 (57.7) | 145 (53.3) | - | 0.391 |
| | | Minor allele | A | 193 (42.3) | 127 (46.7) | - | 0.438 |

CI = 0.9–2.4; p = 0.05), while the heterozygous CT had the lowest risk (OR = 0.6; 95% CI = 0.3–0.9; p = 0.041). Furthermore, no association was found for the C allele and OP phenotype (p = 0.143) (Table 4).

No genotype or allele of *SOST* rs1234612 were found to be significant associated with OP or low BMD (all p>0.05) (Tables 3 and 4).

When analyzing BMD values between genotypes of *VKORC1* rs9934438, no statistical significance was found, except for the heterozygous AG which showed a slight trend of having lower BMD values in the lumbar spine (p = 0.086) (Table 3). The model and allelic analysis revealed no significant statistical associations between groups (all p>0.05) (Table 4).

## Discussion

Even though in the last two decades researchers have continuously searched for the role of genetic factors in the pathogenesis of bone loss, up to date, there is no conclusive etiologic information about OP in this area.

*FDPS* is one of the key-enzymes involved in the mevalonate pathway and it was identified as the main biochemical target of N-BPs [24]. Our research showed a significant association between the presence of the major allele T and osteoporosis (p = 0.005). Also, genotype TT of rs2297480 SNP had significantly lower BMD values in the lumbar spine and total hip (both p<0.05). These results are consistent with the findings of Levy at al. [17], who evaluated the same *FDPS* polymorphism and found a strong association between the presence of the major allele C and low BMD in elderly American women. Marini et al. [18], Olmos et al. [19] and Massart et al [25] evaluated the relation between genotypes of rs2297480 polymorphism and BMD in Caucasian women, but did not find an association with baseline BMD values. The

genotype frequencies are similar to those reported in the American [17], Danish [18], Spanish [19] and Italian [25] population. The current findings are significant for Romanian postmenopausal women, as the T allele and the TT genotype are the most frequent (78% and 60.99%, respectively).

It is known that genes involved in the Wnt pathway are important players in skeletal homeostasis [26]. Polymorphisms in the *LRP5* gene have been previously linked to lumbar spine BMD in adults in multiple studies. Canto-Cetina et al. [27] found a significant association between rs3736228 polymorphism and variations in all BMD sites in Maya-Mestizo women. Interestingly, no significant association of the same SNP with BMD was found in the Slovenian [28], Mexican [29] or Chinese [30] population. In the present research, the CC and CT genotypes of *LRP5* rs3736228 were associated with OP (p = 0.05 and p = 0.041, respectively). Also, postmenopausal women carrying the CC genotype have significantly lower BMD values in the femoral neck and total hip (both p<0.05). Interestingly, in an Italian population, *LRP5* rs3736228 CC genotype tended to have higher BMD values than TT genotype in all BMD sites [25]. And Markatseli et al. [31] reported in a cohort of Greek peri- and postmenopausal women that the presence of CT/TT genotype is consistent with lower lumbar spine BMD. Ezura et al. [32] analyzed rs37362288 genotypes in relation to BMD in adult Japanese women. They found that the homozygous carrying the minor T allele had the lowest BMD scores, and that homozygous carrying the major C allele had the higher BMD. Interestingly, even though the genotypic distribution in the present study is similar to that reported in the Japanese [32], Italian [25] or Greek [31] women, our findings suggest that osteoporosis-risk genotype could be the homozygous CC in postmenopausal Romanian women.

There has been evidence that other Wnt genes, like *SOST*, influence BMD in the general population [33,34]. Valero et al. [35] found that several polymorphisms in the 5' region of the *SOST* gene are associated with BMD in postmenopausal Spanish women. The CTT haplotype at loci rs1234612 was significantly associated with lumbar spine BMD, but not in the femoral neck area. The present study found no association between genotypes and BMD values (all p>0.05). Zhang et al. [36] found that the rs1234612 and the *SOST* haplotype GGTGGATC were associated with adjusted total hip BMD in a large sample of postmenopausal Chinese women. On the contrast, He and al. [37] and Balemans et al. [38] did not find any association between *SOST* gene polymorphisms and BMD in postmenopausal and perimenopausal women, respectively. Velasquez-Cruz et al. [39] concluded that *SOST* polymorphisms contribute to total hip and femoral neck BMD in postmenopausal Mexican-mestizo women. Zhou et al. [36] evaluated several *SOST* polymorphisms regarding treatment response to N-BPs therapy. They found a strong correlation between subjects with homozygous common alleles of rs1234612 and rs851054 and baseline BMD in the lumbar spine. In our study, genotype frequencies of *SOST* rs1234612 were not different than previously reported in the Asian [36,40] population.

There has been evidence that vitamin K plays an important role in maintaining bone strength by acting as a cofactor of the gamma-carboxylase enzyme which activates Vitamin K-dependent proteins in bone, like osteocalcin [41,42]. The *VKORC1* gene is highly polymorphic and mutations in the *VKORC1* gene may modify the gamma-carboxylation of osteocalcin and may influence BMD [20,21]. The effects of *VKORC1* gene on bone have been investigated in several studies. Crawford et al. [43] showed an association of several SNPs in the *VKORC1* gene with BMD in African-Americans. Also, Holzer et al. [44] made a significant association between 3673 G>A genotype and BMD, showing that the homozygous AA had lower BMD values than AG or GG. In the current research, the heterozygous AG tended to have lower BMD values in the lumbar spine area, but it did not reach statistical significance (p = 0.086). Interestingly, Fodor et al. [20] found that the TT genotype of *VKORC1* 1173C>T was the most

frequent in another cohort of postmenopausal Romanian women with OP or osteopenia. Also, a recent study conducted on a Turkish population showed no association between a polymorphism in the *VKORC1* gene and BMD [21]. In our study, the genotype frequencies of the rs9934438 polymorphism were similar to those previously reported in the Romanian [20,22] population. However, relevant studies demonstrating the relationship of this gene to bone biology are still lacking.

This study has several limitations. First, it included only white women and it is unclear if the results can be extrapolated to women of different ethnicity. Secondly, the cohort size was modest, as genetic investigation requires higher number of subjects to achieve sufficient statistical power. And thirdly, the study design was clinic and volunteer based, not population based, therefore potential biases remain.

To our knowledge, this is the first study to detect a positive or negative association between four SNPs in different genes with BMD in a Romanian cohort of postmenopausal women with OP. To date, only one of the SNPs included in this study has been evaluated before in a Romanian population regarding its relation with BMD (*VKORC1* rs9934438) [23]. Also, it is important to mention that the Romanian population is relatively understudied with regards to OP genetics, and the results from this study are potentially of interest for future meta-analyses. Because of the polygenic nature of OP, few genes have major impact on bone metabolism or homeostasis and multiple genes have minor effects, making the classical gene-disease association approach limited, as it mostly lead to inconclusive or controversial results. Regardless of this, data obtained from the current study could be important to Romanian postmenopausal women, as the burden of OP and osteoporosis-related fractures is increasing in Romania, as in other European countries [45].

## Conclusions

Our study showed a strong association between osteoporosis and TT genotype of *FDPS* rs2297480, and with CC genotype of *LRP5* rs3736228. No statistical significance was found between genotypes of *SOST* rs1234612 and *VKORC1* rs9934438 with BMD in postmenopausal Romanian women with osteoporosis.

## Supporting information

**S1 File. Table with genotyping data and bone mineral density.**
(PDF)

## Acknowledgments

The current research derives from the first authors' PhD thesis.

## Author Contributions

**Conceptualization:** Alina Deniza Ciubean, Adrian Pavel Trifa, Anca Dana Buzoianu.

**Data curation:** Alina Deniza Ciubean, Viorela Mihaela Ciortea, Gabriela Bombonica Dogaru, Adrian Pavel Trifa.

**Formal analysis:** Alina Deniza Ciubean, Stefan Cristian Vesa.

**Funding acquisition:** Alina Deniza Ciubean, Rodica Ana Ungur, Viorela Mihaela Ciortea, Ileana Monica Borda, Gabriela Bombonica Dogaru.

**Investigation:** Alina Deniza Ciubean, Adrian Pavel Trifa.

**Methodology:** Alina Deniza Ciubean, Stefan Cristian Vesa, Anca Dana Buzoianu.

**Project administration:** Alina Deniza Ciubean, Viorela Mihaela Ciortea, Gabriela Bombonica Dogaru, Anca Dana Buzoianu.

**Resources:** Alina Deniza Ciubean, Rodica Ana Ungur, Laszlo Irsay, Ileana Monica Borda, Stefan Cristian Vesa.

**Software:** Alina Deniza Ciubean, Ileana Monica Borda, Stefan Cristian Vesa.

**Supervision:** Alina Deniza Ciubean, Laszlo Irsay, Adrian Pavel Trifa, Anca Dana Buzoianu.

**Validation:** Alina Deniza Ciubean, Adrian Pavel Trifa.

**Visualization:** Alina Deniza Ciubean, Laszlo Irsay, Ileana Monica Borda.

**Writing – original draft:** Alina Deniza Ciubean.

**Writing – review & editing:** Alina Deniza Ciubean, Anca Dana Buzoianu.

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
