## [Decision Letter · Decision Letter 0]

21 Oct 2019

PONE-D-19-23734

Polymorphisms of FDPS, LRP5, SOST and VKORC1 genes and their relation with osteoporosis in postmenopausal Romanian women

PLOS ONE

Dear Dr. Ciubean,

Thank you for submitting your manuscript to PLOS ONE. After careful consideration, we feel that it has merit but does not fully meet PLOS ONE’s publication criteria as it currently stands. Therefore, we invite you to submit a revised version of the manuscript that addresses the points raised during the review process.

ACADEMIC EDITOR:  Though the reviewers finds the study is interesting, the authors should address the editorial comments as noted below to further improve the manuscript.  Particularly, they should provide specifics/details and clarify rationale appropriately to study the gene polymorphisms associated with bone anabolic and antiresorptive agents used to treat postmenopausal osteoporosis.

We would appreciate receiving your revised manuscript by Dec 05 2019 11:59PM. To enhance the reproducibility of your results, we recommend that if applicable you deposit your laboratory protocols in protocols.io, where a protocol can be assigned its own identifier (DOI) such that it can be cited independently in the future. For instructions see: http://journals.plos.org/plosone/s/submission-guidelines#loc-laboratory-protocols

We look forward to receiving your revised manuscript.

Kind regards,

Dr. Sakamuri V. Reddy

Academic Editor

PLOS ONE

Journal Requirements:

2. Please provide a sample size and power calculation in the Methods, or discuss the reasons for not performing one before study initiation.

Additional Editor Comments (if provided):

The authors have examined the occurrence of polymorphisms in FDPS, LRP5, SOST and VKORC1 genes with respect to bone mineral density (BMD) in postmenopausal Romanian women. The results showed a strong correlation of BMD and polymorphisms in FDPS (farnesyl diphosphate synthase) gene, but a modest association with LRP5 and SOST genes in postmenopausal Romanian women with osteoporosis and no association with VKORC1. Specific comments to further improve the manuscript are: Introduction-noted very briefly. Please clarify the source and function of SOST and LRP5 genes in bone remodeling with citations appropriately. For ex., a rationale for genes examined ie anti-resorptive and anti-anabolic. Also, clarify SOST gene product Sclerostin protein is primarily produced by the osteocyte has anti-anabolic effect on bone formation. Methods-SNP genotyping is described very briefly, and no details of PCR conditions given. Results-no rationale given for experiments conducted and this section is noted briefly with no details. References-correct ref#29, follow the journal format ex., remove the month, number in refs etc..

Reviewers' comments:

Reviewer's Responses to Questions

**Comments to the Author**

1. Is the manuscript technically sound, and do the data support the conclusions?

Reviewer #1: Yes

2. Has the statistical analysis been performed appropriately and rigorously? 

Reviewer #1: Yes

3. Have the authors made all data underlying the findings in their manuscript fully available?

Reviewer #1: No

4. Is the manuscript presented in an intelligible fashion and written in standard English?

Reviewer #1: Yes

5. Review Comments to the Author

Reviewer #1: This is an interesting report on distribution of 4 osteoporosis genes and their association with osteoporosis in a Romanian women cohort. Overall results are interesting and confirmatory. The paper is well written and contributes to the field of genetics of osteoporosis

6. PLOS authors have the option to publish the peer review history of their article (what does this mean?). If published, this will include your full peer review and any attached files.

Reviewer #1: No

---

## [Author Response · Author response to Decision Letter 0]

11 Nov 2019

Dear Reviewers,

Below you have the responses to the points raised during the review process. 

1. Introduction-noted very briefly. Please clarify the source and function of SOST and LRP5 genes in bone remodeling with citations appropriately. For ex., a rationale for genes examined ie anti-resorptive and anti-anabolic. Also, clarify SOST gene product Sclerostin protein is primarily produced by the osteocyte has anti-anabolic effect on bone formation.

Author response: The introduction has been updated accordingly. 

“ One of the most important signaling pathways in bone in Wnt, which is crucial for bone development during embryogenesis and has a dual role in bone mass regulation, influencing both bone formation and resorption. The components of the Wnt pathway are proteins involved in cell proliferation, differentiation, and apoptosis of bone cells.[5] When cells are stimulated through the membrane receptors low-density lipoprotein receptor-related protein 5/6 (LRP5/6), the architecture of the multiprotein complex is modified, which inhibits b-catenin, leading to its translocation to the nucleus, where it initiates the transcription of target genes.[6] LRP5 is the most important membrane receptor of the Wnt signaling pathway and it was previously tagged in a genome-wide association study to be associated with OP.[7] LRP5 inactivation caused by mutation is responsible for osteoporosis-pseudoglioma syndrome, in which low bone mass and fractures occur.[8] Also, there are several naturally occurring inhibitors of Wnt signaling, such as Dickkopf (DKK) and sclerostin (SOST) proteins that inactivate signaling from LRP5/6 receptors. Sclerostin, encoded by the SOST gene, antagonizes Wnt signaling in both osteocytes and osteoblasts by binding to the LRP5/6 coreceptor and preventing bone formation.[9.10] High-bone-mass syndromes are thought to be caused by inactivating mutations of SOST (sclerosteosis and van Buchem’s disease).[11] Animal studies have indicated that sclerostin inhibition increases bone mass by stimulating bone formation and inhibiting bone resorption.[12]”

2. Methods-SNP genotyping is described very briefly, and no details of PCR conditions given.

Author response: The PCR conditions has been added to the Material and Methods section

“Genomic DNA was obtained from peripheral blood withdrawn on EDTA, using commercially available kits (Quick gDNA MiniPrep kit, Zymo Research, USA; PureLink Genomic DNA Mini Kit, Invitrogen, Thermo Fisher, USA). We genotyped four SNPs (FDPS rs2297480, LRP5 rs3736228, SOST rs1234612, VKORC1 rs9934438) in all patients and controls using the real-time PCR technique. Standard, predesigned TaqMan SNP genotyping assays, containing all the primers and probes needed for genotyping, were purchased from Thermo Fisher (codes C___2737970_10, C__25752205_10, C___7566033_10, C__30204875_10). All the genotyping were performed according to manufacturer’s instructions. The reaction mix contained 10 μl of 2xTaqMan Genotyping Master Mix (Applied Biosystems, Thermo Fisher, USA), 0.5 μl of the corresponding 40xTaqMan SNP genotyping assay, approximately 25 ng of genomic DNA and free-nucleases water to the final volume of 20 μl. The same amplification program was used for all the genotyping, consisting in a pre-read stage of 30 seconds (s) at 60°C, hold stage of 10 minutes (min) at 95°C, followed by the PCR stage, consisting of 40 cycles, each comprising 15 s at 95°C and 1 min at 60°C, and a post-read stage of 30 s at 60°C. All the experiments were run on a QuantStudio 3 real-time PCR machine (Applied Biosystems, Thermo Fisher, USA).”

3. Results-no rationale given for experiments conducted and this section is noted briefly with no details.

Author response: The rationale behind the experiments have been added

” After dividing the patients into groups, analysis of the codominant model was performed. Codominant models hypothesize that the major allele homozygotes, the heterozygotes or the minor allele homozygotes are associated with the lowest, the intermediate, and the highest risk, respectively, or they are associated with the highest, the intermediate, and the lowest risk, respectively.”

4. References-correct ref#29, follow the journal format ex., remove the month, number in refs etc..

Author response: all references have been modified accordingly.

5. Have the authors made all data underlying the findings in their manuscript fully available? Reviewer response: No

 Author response: An Excel file with the raw data has been uploaded with the manuscript files

 In the name of all the authors, we thank you for the peer-review process, and we hope the modifications are sufficient and the article is ready to be published in its currents form. If any further clarifications are required, we are happy to give them. 

Sincerely,

Dr. Alina Deniza Ciubean

---

## [Editor Report · Decision Letter 1]

13 Nov 2019

Polymorphisms of FDPS, LRP5, SOST and VKORC1 genes and their relation with osteoporosis in postmenopausal Romanian women

PONE-D-19-23734R1

Dear Dr. Ciubean,

We are pleased to inform you that your manuscript has been judged scientifically suitable for publication and will be formally accepted for publication once it complies with all outstanding technical requirements.

With kind regards,

Dr. Sakamuri V. Reddy

Academic Editor

PLOS ONE
---

## [Editor Report · Acceptance letter]

19 Nov 2019

PONE-D-19-23734R1 

Polymorphisms of FDPS, LRP5, SOST and VKORC1 genes and their relation with osteoporosis in postmenopausal Romanian women 

Dear Dr. Ciubean:

I am pleased to inform you that your manuscript has been deemed suitable for publication in PLOS ONE. Congratulations! Your manuscript is now with our production department. 

With kind regards,

on behalf of

Dr. Sakamuri V. Reddy 

Academic Editor

PLOS ONE